# Can One Embedding Fit All? A Multi-Interest Learning Paradigm Towards Improving User Interest Diversity Fairness

## ABSTRACT

Recommender systems have gained widespread applications across various domains owing to their superior ability to understand and capture users' interests. However, the complexity and nuanced nature of users' interests, which can span a wide range of diversity, pose a significant challenge in delivering fair recommendations. In real-world scenarios, user preferences vary significantly; some users show a clear preference toward certain item categories, while others have a broad interest in diverse ones. Even though it is expected that all users should receive high-quality recommendations, the effectiveness of recommender systems in catering to this disparate interest diversity remains under-explored.

In this work, we investigate *whether users in different groups with varied levels of interest diversity are treated fairly.* Our empirical experiments reveal an inherent disparity: users who have a wider range of interests often receive lower-quality recommendations. To achieve fairer recommendations, we propose a multi-interest framework that uses multiple (virtual) interest embeddings, rather than the utilization of single embedding to represent individual users. Specifically, the framework consists of stacked multi-interest representation layers. Each layer includes an interest embedding generator that derives virtual interests from globally shared interest parameters, and a center embedding aggregator that facilitates multi-hop aggregation. The experiments have demonstrated the effectiveness of the proposed method in achieving better trade-off between fairness and utility across various datasets and backbones. Our code and datasets are available at: Code.

## KEYWORDS

Fairness, Diversity, Multi-Interest Recommendations

**ACM Reference Format:**

Anonymous Author(s). 2023. Can One Embedding Fit All? A Multi-Interest Learning Paradigm Towards Improving User Interest Diversity Fairness. In *Proceedings of ACM Conference (Conference'17).* ACM, New York, NY, USA, 10 pages. https://doi.org/10.1145/nnnnnnn.nnnnnnn

## 1 INTRODUCTION

Recommender systems (RSs) have been widely applied in different domains, such as news recommendation [17], friend recommendation [7], etc. While a plethora of RSs have been proposed [12, 22, 29],

*Conference'17, July 2017, Washington, DC, USA*
© 2023 Association for Computing Machinery.
ACM ISBN 978-x-xxxx-xxxx-x/YY/MM...$15.00
https://doi.org/10.1145/nnnnnnn.nnnnnnn

(A) Food Recommendation    (B) Dating Recommendation

Specific Interests  Broad Interests    Specific Interests  Broad Interests

**Figure 1: Why diverse interests matter? Two real-world examples of RS applications (A) Food recommendation (B) Dating recommendation[1].**

the main focus is on maximizing the overall utility, typically measured by metrics like Recall, F1, and NDCG [1]. These metrics offer a comprehensive view on the accuracy of recommendations and the system's ability in capturing user interests. However, solely relying on these utility-based measurements can cause issues: (1) it hides biases across distinct user groups, posing fairness concerns; and (2) it overshadows underlying performance bottlenecks, impeding potential utility enhancements. In light of these issues, recent studies have adopted a group-centric lens for recommendations [15, 31, 32]. Investigations have been conducted on user groups defined by explicit attributes (i.e., sensitive features) such as gender [32], race [35], as well as implicit features (i.e., extracted from interactions) such as number of interactions and amount of purchases [15, 31]. These studies highlight group-specific biases and advocate for solutions that ensure fairness. Given the rich existing literature focused on explicit sensitive attributes, our study dives into the implicit features and specifically focuses on a novel perspective termed *user interest diversity.* We investigate the following research question:

> **Are users of varied interest diversity treated fairly in RSs?**

*Firstly,* imbalanced user satisfaction could undermine the overall utility of the platform and even result in dissatisfied users leaving (i.e., increased user defection) [15, 31]. For example, in the context of food recommendation in Fig. 1(A), some users prefer a limited number of cuisines while others have more flexible tastes. Satisfying all users is a primary goal. *Secondly,* if the platform fails to equitably accommodate these diverse preferences, it not only raises issues of user satisfaction but also poses significant ethical concerns. Online dating recommendation in Fig. 1(B) serves as a pertinent example. Users exhibit a spectrum of sexual orientations, including homosexuality, bisexuality, heterosexuality, and more. While homosexual and heterosexual users have more specific preferences related to gender interests, bisexual users might exhibit a broader range of interests. Ensuring a fair and unbiased system for users with varied interest diversity is a core requirement for ethical consideration.

---

[1]Note that the illustration does not represent the authors' perspective on the concept of binary genders.

To explore the fairness of existing models towards users exhibiting various levels of interest diversity, we conduct a preliminary experiment with detailed analysis in Sec. 2. In particular, we contemplate two scenarios: one where item category information (e.g., movie genres) is available, and another where it is not. We then define two interest diversity metrics. Following this, we categorize users into groups based on interest diversity and compare the utility metrics of the recommendations they receive. The results reveal a pattern that users with higher interest diversity tend to receive lower recommendation performance. This observation remains consistent across multiple datasets, models, definitions of interest diversity, and group partitions. Our experiments indicates that the unfairness among user groups with varied interest diversity (i.e., *user interest diversity unfairness*) indeed exists. To alleviate such unfairness without compromising the overall utility performance, it's necessary to enhance the recommendations for users with high interest diversity, as this is the system's performance bottleneck. We explore the cause of performance disparity among user groups, and our conclusion aligns with prior work [2, 33], which suggests that a single embedding is insufficient to capture users' interests.

To this end, we propose a multi-interest framework to improve user interest diversity fairness, that can be integrated into existing RS models. In our multi-interest framework, each user is composed of a center embedding representing users' main characteristic and multiple virtual embeddings, reflecting users' interests derived from their interacted items. We develop multi-interest representation layers to learn better user embeddings, especially for users with high interest diversity. Each layer includes an interest embedding generator that derives virtual interest embeddings from globally shared interest parameters, and a center embedding aggregator that facilitates multi-hop aggregation. As such, the designed mechanism can automatically assign different interest numbers that are generally consistent with the interest diversity in an implicit manner. Experimental results validate the effectiveness of our framework in achieving a better trade-off between fairness and utility performance. Our main contributions are summarized as follows:

- **Consistent Disparity Identification**: We identify the unfair treatment among users with varied levels of interest diversity, where users with broader interests tend to receive lower recommendation quality. This pattern has been empirically verified to be consistent across multiple datasets, models, diversity metrics, and group partitions.

- **Multi-interest Framework Design**: We delve into the potential reason causing the disparity from the embedding space where we observe the insufficiency of using single embedding to represent users and items due to their complex multi-faceted interactions. This motivates us to propose a multi-interest framework which is both model-agnostic and parameter-efficient.

- **Better Fairness-Utility Tradeoff**: Our proposed multi-interest framework outperforms the backbone models and fairness baselines by achieving the optimal balance between fairness and utility. Also, it offers superior and more balanced embedding alignment, along with more diverse recommendations.

**Table 1: Notations.**

| Notations | Descriptions |
|---|---|
| $\mathcal{I}_u$ | User $u$'s interactions $\mathcal{I}_u = [i_1, i_2, ..., i_{d_u}]$ |
| $d_u$ | Number of interactions |
| $C_u$ | Category set of $\mathcal{I}_u$ |
| $N_u^c$ | Number of user $u$'s interaction in category $c$ |
| $\mathrm{D_{cat}}/\mathrm{D_{emb}}$ | User interest diversity via item category/embedding |
| $\phi(\cdot, \cdot)$ | Similarity function |
| $\mathbf{e}_u/\mathbf{e}_i$ | User/Item embeddings |
| $\tilde{\mathbf{e}}_u/\tilde{\mathbf{e}}_i$ | Normalized User/Item embeddings |
| $\mathbf{A}$ | Adjacency matrix |
| $\mathbf{D}$ | Degree matrix |
| $K/k$ | Number of interests/$k$-th interest |
| $N$ | Number of users and items |
| $d$ | Embedding dimension |
| $\hat{y}_{ui}$ | Relevance score between user $u$ and item $i$ |
| $\mathcal{N}_{v_i}$ | The neighborhood set of node $v_i$ |
| $\mathbf{E}_C^l$ | Center embeddings at layer $l$ |
| $\mathbf{E}_V^l$ | Virtual interest embeddings at layer $l$ |
| $\mathbf{w}_k^l$ | Global interest parameter of $k$-th interest at layer $l$ |
| $\downarrow/\uparrow$ | The lower/higher the better |

## 2 USER INTEREST DIVERSITY UNFAIRNESS

In this section, we investigate how existing RSs treat users with varied levels of interest diversity. First, we formally define interest diversity, concatering two scenarios where item category is available or not. Then, we categorize users into groups with varied levels of interest diversity. Ultimately, we demonstrate the performance across different groups using two representative recommendation models: LightGCN [12] and CAGCN* [29]. The disparate group performance reveals the existence of user interest diversity unfairness. Notations used in the paper are summarized in Table 1.

### 2.1 Interest Diversity Definition

User interest diversity aims to measure the dissimilarity of the items interacted with each user in the training data (i.e., users' historical interactions). Based on whether category information is available, we define interest diversity based on item category or item embedding.

*Definition 2.1.* **Interest Diversity via Item Category.** Given user $u$'s historical interaction $\mathcal{I}_u = [i_1, i_2, ..., i_{d_u}]$ where $d_u$ is the number of interactions and $C_u$ is the set of categories of items user $u$ has interacted with, $N_u^c$ denotes the number of items from user $u$'s interaction belonging to category $c$, we define user $u$'s interest diversity $\mathrm{D_{cat}}(u)$ following Simpson's Index of Diversity [24]:

$$\mathrm{D_{cat}}(u) = 1 - \frac{\sum_{c \in C_u} N_u^c (N_u^c - 1)}{|\mathcal{I}_u|(|\mathcal{I}_u| - 1)}. \tag{1}$$

*Definition 2.2.* **Interest Diversity via Item Embedding.** Given the pretrained item embeddings, user $u$'s interest diversity $\mathrm{D_{emb}}(u)$ is as follows:

$$\mathrm{D_{emb}}(u) = 1 - \mathbb{E}_{(i,i') \in \mathcal{I}_u \times \mathcal{I}_u} \phi(\mathbf{e}_i, \mathbf{e}_{i'}), \tag{2}$$

where $\phi(\mathbf{e}_i, \mathbf{e}_{i'}) = \frac{\mathbf{e}_i \cdot \mathbf{e}_{i'}}{\|\mathbf{e}_i\| \|\mathbf{e}_{i'}\|}$ is the cosine similarity between the embeddings of two items $i, i'$.

Essentially, $\mathrm{D_{cat}}(u)$ measures the probability that two randomly sampled items are from different categories, and $\mathrm{D_{emb}}(u)$ measures the dissimilarity between the interacted items in their embedding space. For both scenarios, a larger value indicates a higher level of interest diversity. Unless specified, we use $\mathrm{D_{cat}}$ for default.

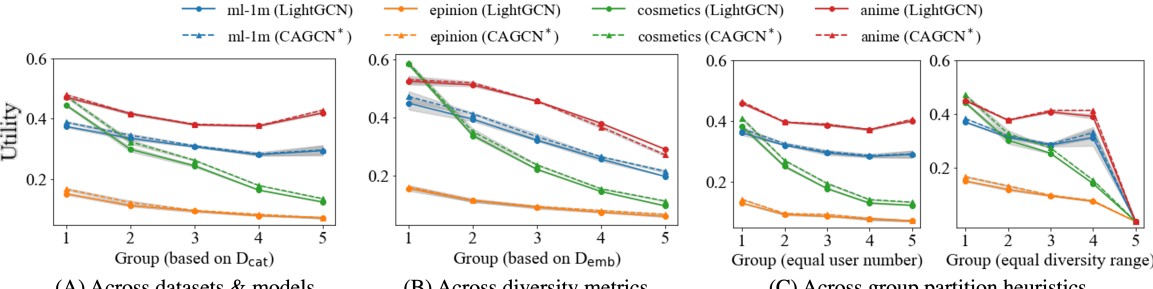

Figure 2: Group recommendation performance (Recall ↑): the pattern that *users with more diverse interests generally receive lower recommendation quality* is consistent across various *datasets, models, diversity metrics,* and *group partitions*. A larger group ID indicates a higher level of user interest diversity.

## 2.2 Group Partition

Given users' interest diversity, we group users with k-means clustering [18]. The number of clusters is determined using the commonly-used elbow method [25]. The assignment of clusters subsequently defines the group partition, with a higher group ID indicating a higher diversity of interests. It's worth noting that there are alternative methods to group users, e.g., dividing users into equal sized groups based on number of users, or range of user interest diversity. Unless specified otherwise, we primarily rely on k-means clustering in the experiments.

## 2.3 Preliminary Results

Given the exceptional performance of utilizing graphs in RSs, we select two graph-based models for evaluation: LightGCN [12] and CAGCN* [29]. The former is a widely recognized and frequently used model. The latter is a newer development and improves the overall utility by reducing the emphasis of neighbors not adhering to the main interest which is closely related to our topic.

We evaluate them on four datasets including ml-1m, epinion, embmetics, and anime, the details of which will be described in Sec. 4.1.1. The preliminary results across different scenarios are illustrated in Fig. 2. Specifically, Fig. 2(A) is the group utility performance (Recall) where groups are divided based on k-means clustering with $D_{cat}$ as the diversity metric. The curves suggest a trend that as interest diversity increases, the group utility performance generally decreases. This pattern is observable across multiple datasets and models. We also explore another diversity definition $D_{emb}$ in Fig. 2(B) which shows a similar trend. Additionally, we obtain results based on different group partitions including the equal user number and equal user interest diversity range in Fig. 2(C). The results show a consistent trend across various datasets, models, diversity metrics, and group partitions that users with diverse interests generally receive a lower recommendation quality. This indicates the existence of user interest diversity unfairness, which jeopardizes the user experience for user with diverse interests.

## 3 THE MULTI-INTEREST FRAMEWORK

To mitigate the user interest diversity unfairness identified in Sec. 2, we dive into the source of unfairness from the alignment and misalignment between user and item embeddings. Our empirical findings indicate a trend in alignment that correlates with the observed performance disparities: user group with diverse interests has poor

performance as well as poor alignment. We hypothesize that the suboptimal alignment arises from the inadequacy of using single embedding to align user's diverse interests with the interacted items (illustrated in Fig. 4). To improve the alignment, especially for users with higher level of interest diversity, we propose a multi-interest framework where each user is represented by multiple (virtual) interest embeddings. Based on the proposed framework, we improve the alignment for users with high interest diversity, thereby improving their recommendation performance and alleviating the detected performance bias.

Next, we discuss the source of unfairness in Sec. 3.1, give an overview of the multi-interest framework in Sec. 3.2, elaborate the components details in Sec. 3.3 and the optimization in Sec. 3.4.

## 3.1 Source of Unfairness

Since the core component in majority RSs is to learn high-quality user and item embeddings, we investigate the root cause of *user interest diversity unfairness* from the embedding space. Prior research has underscored the correlation between embedding alignment (i.e., the capacity to bring users and their associated items closer in the embedding space) and utility performance [26, 27]. A superior alignment typically correlates with a better performance. The alignment definition is as follows:

$$\text{Alignment} = \mathbb{E}_{(u,i) \sim p_{\text{pos}}} \| \tilde{\mathbf{e}}_u - \tilde{\mathbf{e}}_i \|^2, \tag{3}$$

where $\tilde{\mathbf{e}}_u$ and $\tilde{\mathbf{e}}_i$ are the $l_2$ normalized user and item embeddings from historical interacted pairs. It measures the Euclidean distance in the unit hypersphere and a lower Alignment score (aka. shorter distance) corresponds to better utility performance. To uncover the potential reason for unfair recommendation performance across different user groups, we measure the average Alignment in each group. Results on ml-1m in Fig. 3 (results for other datasets are included in Appedix A) show that (1) CAGCN* exhibits superior alignment compared to LightGCN, a consistency mirrored in the performance illustrated in Fig. 2; (2) Users displaying a broader spectrum of interests tend to have larger Alignment scores in the embedding space. This suggests that the current recommendation models are not effective in aligning users and items, particularly when users have a wide array of interests.

Fig.4(A) depicts the alignment challenge for user with high interest diversity. When the user is represented by a single embedding, to achieve an optimal alignment with every interacted item, the learned single embedding falls in-between the interacted items.

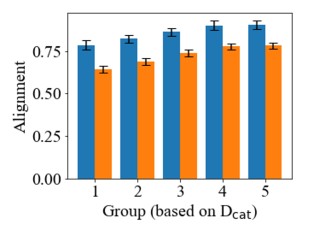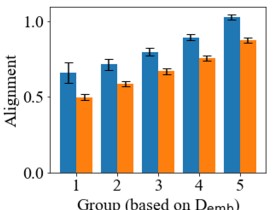

Figure 3: Group-level embedding alignment (↓) of ml-1m dataset based on LightGCN and CAGCN*.

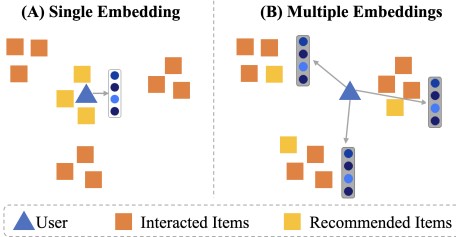

Figure 4: Multi-interest motivation: single embedding is insufficient to capture users' diverse interests.

This results in a poor alignment with the real interests. Such insufficiency of using single embedding to align interacted items, that are from diverse interests, motivates us to use multiple embeddings to represent different user interests [2, 14, 33]. As shown in Fig. 4(B), the user has multiple embeddings. For items belonging to diverse interests, the embeddings can be automatically obtained and they have a better alignment with the corresponding interacted items in the embedding space. Comparing the scenarios of single-interest and multi-interest, we find that owing to a better alignment, the recommended items in Fig. 4(B) are more accurate than Fig. 4(A). This underscores the potential of the multi-interest approach.

## 3.2 Model Architecture

Fig. 5 shows the multi-interest framework where each user/item has different types of embeddings, including (1) center embeddings $\mathbf{E}_C^l \in \mathbb{R}^{N \times d}$ representing users/items main characteristic/features where $N$ is the number of node (including users and items) and $d$ is the dimension; (2) interest (virtual) embeddings $\mathbf{E}_V^l \in \mathbb{R}^{N \times K \times d}$ which relate to specific interests where $K$ is the number of interests (for simplicity, we denote $\mathbf{E}_V$ as virtual embeddings hereafter). Among these embeddings, center embeddings are learnable parameters while the virtual embeddings are calculated based on center embeddings via attentions. This mechanism avoids introducing a large number of learnable parameters by sharing the global interest parameters $\mathbf{w}_k^l$ in the attention mechanism. We represent the $k$-th virtual embedding of node $v_u$ as $\mathbf{E}_V^L[v_u, k]$ and the user center embedding as $\mathbf{E}_C^L[v_u]$. Similar notations apply to the item side.

With these notations, the framework is as follows: (1) Given the user-item bipartite graph, user and item embeddings are obtained through the multi-interest representation layers (details in Sec. 3.3); (2) After obtaining the embeddings, the relevance score $\hat{y}_{ui}$ for user, item pair $(v_u, v_i)$ is calculated based on the last layer representations where L is the number of hops:

$$\hat{y}_{ui} = \max_{k=1}^{K} \mathbf{E}_V^L[v_u, k]^\top \mathbf{E}_C^L[v_i] + \max_{k=1}^{K} \mathbf{E}_V^L[v_i, k]^\top \mathbf{E}_C^L[v_u]; \quad (4)$$

(3) These predicted relevance scores are optimized via Bayesian Personalized Ranking Loss (BPR) loss [22] $\mathcal{L}_{\mathrm{BPR}}$.

Note that the relevance score in Eq.(4) is different from the calculation in previous recommendation models [12, 29] or multi-interest-based session recommendation [2, 33]. In previous works, because user and item only have single embeddings, the dot product between the user and the item embedding (i.e., $\mathbf{e}_u^\top \mathbf{e}_i$) denotes their relevance score. In multi-interest based session recommendation,

only items have learnable parameters and users/sessions are calculated based on items ($K$ embeddings with $\mathbf{e}_u^k$ denoting the $k$-th interest) and therefore $\max_{k=1}^{K} \mathbf{e}_u^{k\top} \mathbf{e}_i$ is sufficient to update the item embeddings. However, similar to LightGCN, we have both user and item embeddings to learn. Simply optimizing the traditional multi-interest relevance score that is commonly used in session-based representation cannot utilize user embeddings, indicating it is not suitable in our case. Therefore, we use the symmetric scores shown in Eq.(4) where both user and item embeddings are optimized.

## 3.3 Multi-Interest Representation Layer

Next, we introduce the details of multi-interest representation layer, which is at the core of the architecture and designed to learn, calculate and aggregate multiple embeddings. The model is composed of stacked layers to deliver the final user and item embeddings.

*3.3.1 Interest embedding generation:* Virtual embeddings of $l$-th layer for node $v$ and $k$-th interest (i.e., $\mathbf{E}_V^l[v, k]$) is calculated in Eq.(5) as the weighted average of the center embeddings of neighbors. The weight is calculated in Eq.(6) based on Softmax attention mechanism where $T$ is the temperature to control the Softmax smoothness. The input logits to Softmax function are cosine distances between virtual embeddings and the global interest $\mathbf{w}_k^l$. Intuitively, if an item is related to the $k$-th interest, the attention will be higher and lead to larger contribution to the aggregates from this item. Therefore, $\mathbf{E}_V^l[v, k]$ captures information related to $k$-th interest.

$$\mathbf{E}_V^l[v, k] = \sum_{v_n \in \mathcal{N}_v} a_{k, v_n}^l \mathbf{E}_C^l[v] \quad (5)$$

$$a_{k, v_n}^l = \frac{\exp(\phi(\mathbf{E}_C^l[v_n], \mathbf{w}_k^l)/T)}{\sum_i \exp(\phi(\mathbf{E}_C^l[v_n], \mathbf{w}_i^l)/T)} \quad (6)$$

*3.3.2 Center embedding aggregator:* We adopt the similar approach as LightGCN [12] to update embeddings based on topology of the graph. Different from LightGCN, we use virtual embeddings to update the center embedding as in Eq.(7). Since virtual embeddings have extra dimension in interest, these embeddings need to be transformed to the same dimension as center embedding before the aggregation. We use an *argmax* operator to select the interest id of the "matching slide" called *mid*. The embeddings of *mid* index has the highest dot product similarity with the node's center embedding. Such operator has been commonly used in multi-interest literature [2, 33] and has been verified to have faster convergence and better performance compared with other ways to use multi-interests [14]. In detail, for each node $v$ whose center embedding is

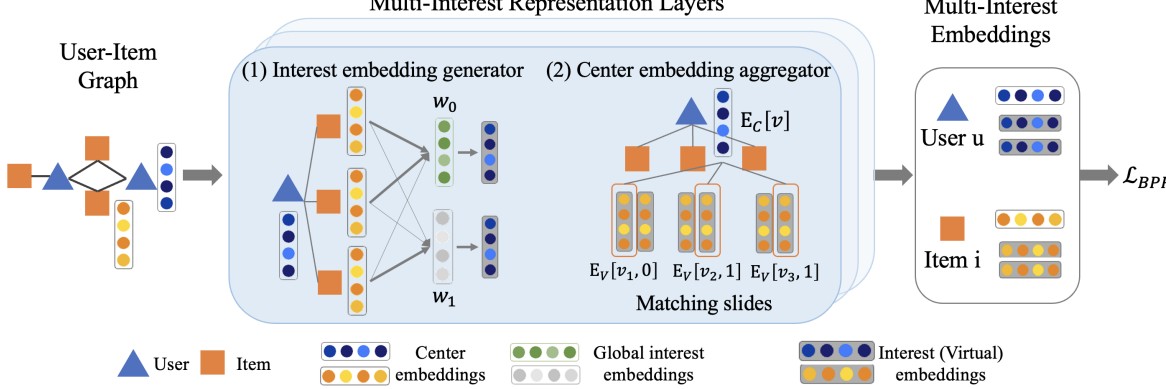

Figure 5: Multi-interest framework (interest number equals two): rather than a single embedding, each user/item is represented by multiple embeddings (i.e., center and virtual). Center embeddings and global interest embeddings are learnable parameters while the interest (virtual) embeddings are calulated without assigning extra parameters.

$E_C^l[v]$, the id of the matching slide for one neighbor node $v_n \in \mathcal{N}_v$ is selected as:

$$\text{mid}(v, v_n, l) = \text{argmax}_{k=1}^K (E_V^l[v_n, k]^\top E_C^l[v])$$

Given the "matching slide," the aggregation process is as follows:

$$E_C^{l+1}[v] = \sum_{v_n \in \mathcal{N}_v} \frac{1}{\sqrt{d_v d_{v_n}}} E_V^l[v_n, \text{mid}(v, v_n, l)] \qquad (7)$$

## 3.4 Optimization

We utilize the BPR loss [22] ($\mathcal{L}_{\text{BPR}}$) to train our multi-interest RS.

$$\mathcal{L}_{\text{BPR}} = - \sum_{(u,i,j) \in \mathcal{D}} \log \sigma(\hat{y}_{ui} - \hat{y}_{uj}) + \lambda_\Theta \|\Theta\|^2,$$

where $\mathcal{D} = \{(u, i, j) | u \in \mathcal{U} \wedge i \in \mathcal{I}_u^+ \wedge j \in \mathcal{I}_u^-\}$ is the training dataset and $\mathcal{U}$ is the total user set, $\mathcal{I}_u^+/\mathcal{I}_u^-$ are the item sets that user $u$ has/hasn't interacted with. $\sigma(\cdot)$ is Sigmoid function. $\Theta$ denotes the model parameter with $\lambda_\Theta$ controlling the $L_2$ norm regulation to prevent over-fitting. $\hat{y}_{ui}$ is the predicted preference/relevance score computed based on Eq.(4).

## 4 EXPERIMENTS

In this section, we evaluate the performance of our multi-interest framework on on real-world datasets and compare the utility and fairness performance with various representative methods. Through experiments, we aim to answer the following research questions:

- **RQ1:** Does our proposed multi-interest framework achieve a better utility-fairness trade-off than the baseline methods?
- **RQ2:** Is the multi-interest framework able to learn higher-quality embeddings with better alignment?
- **RQ3:** Can the multi-interest framework learn to match the number of users' interest embeddings with the diversity of their historical interactions?
- **RQ4:** Can the multi-interest framework provide extra benefits beyond accuracy and fairness, e.g., recommendation diversity?
- **RQ5:** How do the hyperparameters affect the performance?

Table 2: Dataset statistics.

| Dataset | # Edges | # Users | # Items | # Category |
|---|---|---|---|---|
| ml-1m | 223305 | 5645 | 2357 | 18 |
| epinion | 163320 | 11875 | 11164 | 26 |
| cosmetics | 930275 | 53238 | 28310 | 400 |
| anime | 901328 | 40112 | 4514 | 76 |

## 4.1 Experimental Setup

*4.1.1 Datasets.* We evaluate the proposed multi-interest framework on four datasets including ml-1m, epinion, cosmetics, and anime[2]. We pre-process data by (1) filtering edges by maintaining the highest rating score so that the remaining edges show strong preferences; and (2) applying k-core filtering iteratively to remove users with interaction number smaller than 5. After that, we randomly split the dataset into train/validation/test based on 60%/20%/20% proportions. The statistics of the pre-processed datasets are summarized in Table 2.

*4.1.2 Baselines.* To verify whether our framework can achieve a better trade-off between fairness and utility, and further generalize to different backbones, we compare the performance of two representative recommendation backbones (LightGCN [12] and CAGCN* [29]) before/after equipping our proposed multi-interest framework. For a fair comparison, we also apply other fair baselines to the backbones including DRO [11] and ARL [13]. Note that all these methods are group-agnostic which means that the group partition is unavailable during training. The descriptions for the compared methods are as follows:

- **LightGCN** [12] is a GNN-based method that aggregates high-order neighborhood information and simplifies traditional GCN by removing the linear transformation and nonlinear activation.
- **CAGCN*** [29] is a fusion model of LightGCN [12] and Collaboration-Aware Graph Convolutional Network (CAGCN [29]). It analyzes how message-passing captures collaborative filtering (CF) effect and pre-computes a topological metric, Common Interacted Ratio (CIR), for collaboration-aware propagation.

---

[2]Datasets are available at: ml-1m, epinion, cosmetics, anime

**Table 3: Fairness and utility performance (The best is highlighted in bold and the runner-up is underlined).**

| Backbone | Method | ml-1m | | epinion | | cosmetics | | anime | | Avg Rank |
|---|---|---|---|---|---|---|---|---|---|---|
| | | Recall↑ | Unfairness↓ | Recall↑ | Unfairness↓ | Recall↑ | Unfairness↓ | Recall↑ | Unfairness↓ | |
| **LightGCN** | - | 0.3087 | **0.0376**/0.1018 | 0.0904 | 0.0320/0.0378 | 0.2116 | 0.1260/0.1942 | 0.4015 | **0.0384**/0.098 | 2.08 |
| | DRO | **0.3143** | 0.0409/0.1047 | **0.0926** | 0.0377/0.0.451 | 0.2104 | 0.1296/0.2013 | 0.4029 | 0.0453/0.1102 | 2.92 |
| | ARL | 0.2973 | **0.0376**/0.1058 | 0.0850 | **0.0316**/0.0381 | 0.1941 | 0.1199/0.1730 | 0.3844 | 0.0407/0.1110 | 2.83 |
| | Multi | 0.3116 | 0.0385/**0.0856** | 0.0901 | 0.0364/**0.0222** | **0.2405** | **0.1193**/**0.1494** | **0.4239** | 0.0430/0.1136 | **1.92** |
| **CAGCN*** | - | 0.3141 | 0.0429/0.1054 | **0.0948** | 0.0380/0.0373 | 0.2286 | 0.1332/0.1903 | 0.4044 | 0.0415/0.1096 | 2.83 |
| | DRO | **0.3173** | 0.0401/0.0946 | 0.0927 | 0.0383/0.0375 | 0.2294 | 0.1350/0.1891 | 0.4024 | 0.0414/0.1012 | 2.58 |
| | ARL | 0.3024 | **0.0367**/0.1121 | 0.0912 | **0.0354**/0.0380 | 0.2167 | **0.1257**/0.1734 | 0.3884 | **0.0413**/0.0985 | 2.67 |
| | Multi | 0.3107 | 0.0411/**0.0921** | 0.0922 | 0.0378/**0.0212** | **0.2548** | 0.1297/**0.1368** | 0.4237 | 0.0417/**0.0904** | **1.92** |

- **DRO** [11] is a group-agnostic optimization approach that aims to improve the performance of the worst-case instances via distributionally robust learning.
- **ARL** [13] is a group-agnostic optimization approach that leverages an adversary module to automatically adjust the weight in the training loss so that instances with higher loss will be assigned higher weights.
- **Multi** is the multi-interest framework proposed in this paper. It learns multiple interest embeddings to represent each user to mitigate the performance gap among user groups.

*4.1.3 Implementation details.* For all methods, we use Adam optimizer for training and set the learning rate to 0.001, batch size to 2048, L2 coefficient to 0.001, and embedding dimension to 32. We early stop the training process when the best validation score remains unchanged for 25 epochs. Trend coefficient in CAGCN* is set to 1.0. Temperature in the Softmax function is set to 2.0. The model hyperparameters are selected based on the best recall value during validation. For each model, we tune the number of hops within $\{1, 2, 3\}$. Additionally, for DRO-based model, we tune the hyperparameter $\eta$ within $\{0.0, 0.2, 0.4, 0.6, 0.8, 1.0\}$. For our model, we tune interest number within $\{2, 4, 8, 16\}$. We run the experiments three times and report the average results. The best hyperparameters for each model are reported in Appendix B. When applied on CAGCN* backbone, the aggregation weights in Eq.(7) is substituted with the pre-computed topological-based weights introduced in work [29].

*4.1.4 Metrics.* For utility performance, we adopt Recall@20 and NDCG@20. For fairness performance, we use the standard deviation of the utility performance across user groups. The deviation measures the performance gap among groups, and a larger score signifies lower fairness. Based on group partitions via interest diversity metrics $D_{cat}$ and $D_{emb}$, we report two corresponding (un)fairness scores. This setting can evaluate whether the group-agnostic models are effective for different group partitions.

## 4.2 Performance Comparison (RQ1)

We present the utility and fairness scores for LightGCN and CAGCN* backbones respectively in Table 3 (The results based on another utility metric NDCG is included in Appendix C). Since the standard deviations for all methods across various seeds are negligible compared with the main performance, we leave them out. From the result, we draw several observations:

- *The multi-interest framework has the best fairness-utility trade-off in general.* Our proposed method achieves the best and runner up performance in most of the times when compared with other methods. Upon calculating the average rank for each method, ours emerges as the leader in both backbones. While the current rank of 1.92 indicates some room for enhancement towards the optimal rank of 1, it underscores the efficacy and potential of the multi-interest framework in balancing fairness and utility.
- *The multi-interest framework works better with large dataset.* In cosmetics dataset, which has the highest count of items and categories, our method consistently delivers enhanced performance in both fairness and utility. Given the diversity of items and categories, learning varied interests becomes more essential, amplifying the advantages.
- *The multi-interest framework is more stable across backbones compared with other fairness baselines (i.e., DRO and ARL).* DRO and ARL rank higher than the base model CAGCN*, however, their rank drops when integrated into LightGCN. This underscores the complexity of maintaining an optimal balance across different models. Furthermore, such distinct performance variations of DRO and ARL across different backbones can be attributed to their inherent design. These methods were specifically designed to enhance the performance for instances with suboptimal recommendations. While Fig. 2 demonstrates that the user group with diverse interests has the poorest average performance and is expected to gain the most, other factors, such as the percentage of under-performing users in each group, play a role. If other groups have a higher proportion of users with poor recommendations, they might obtain greater benefits, thereby increasing the unfairness. Therefore, we can observe in some cases (e.g., DRO in ml-1m with LightGCN backbone) that the utility improves and the fairness drops. Such percentage in each group can vary across models, resulting in high instability of DRO and ARL due to their heavy reliance on the performance distribution. This suggests that DRO and ARL are not universally effective in the current context. In contrast, the multi-interest framework relies on the underlying interests rather than performance, which is more closely related to the current setting and more stable.

## 4.3 Representation Quality (RQ2)

Multiple embeddings are expected to learn a better embedding distribution compared with single embeddings (Fig. 4), especially for the embedding alignment between user and interacted items. To

Table 4: Embedding alignment (Results with improved alignment compared with backbone are highlighted in bold).

| Method | ml-1m | epinion | cosmetics | anime |
|---|---|---|---|---|
| LightGCN | 0.8774 | 0.5951 | 0.7937 | 1.0165 |
| Multi-LightGCN | **0.5007** | **0.4111** | **0.5396** | **0.7514** |
| CAGCN* | 0.7512 | 0.5429 | 0.7069 | 0.9118 |
| Multi-CAGCN* | **0.4315** | **0.2973** | **0.4694** | **0.7176** |

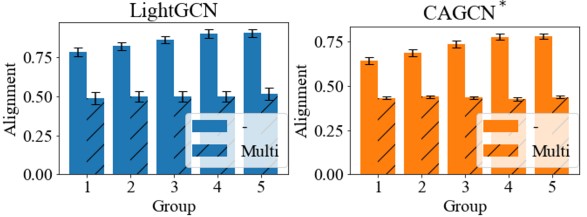

Figure 6: Group-level embedding alignment of ml-1m dataset based on two backbones.

evaluate this, we calculate the average alignment based on the backbones and their multi-version. Table 4 shows that multi-interest improves the alignment consistently. This suggests that the framework effectively brings users and their interacted items closer in the embedding space. However, an intriguing observation arises when examining performance metrics. While the improved alignment in CAGCN* leads to superior utility performance compared to Light-GCN in Table 3, the enhanced alignment in the multi-version does not always result in better utility performance relative to the backbones. This inconsistency may arise from the trade-off between alignment and uniformity [26]. Specifically, while alignment improves, it could lead to reduced uniformity in the multi-version due to more user embeddings, which offsets the anticipated enhancements. The nuanced interplay between alignment and uniformity, and strategies to effectively balance them, present intriguing avenues for future exploration in multi-interest scenario.

Beyond evaluating overall alignment, we delve into embedding alignment at the group level. In Fig. 6, there's a discernible trend when comparing the backbone to its multi-version: alignment appears more evenly distributed across different groups. Since alignment is closely related to the utility performance, it contributes to a fair recommendation across groups, which follows our expectation.

## 4.4 Interest Matching (RQ3)

For each user, our multi-interest framework initially assigns the same number of interests (i.e., $K$). Given the underlying assumption that users exhibit varied levels of interest diversity, can the model autonomously adjust the number of interests even if it begins with an equal allocation? To answer this question, we obtain the set of interests that matches the recommended items (i.e., for each item, the matched interest is the specific interest that has the maximum relevance score) and calculate the average matched interest number for each group. Results in Fig. 7 show that for the first three groups, users with more diverse interests have been assigned a larger interest number, indicating that our model has the ability to distinguish different interest diversity and can automatically cater

Table 5: Diversity measured by $D_{cat}$ and $D_{emb}$ (Results with improved diversity compared with backbone are in bold).

| Diversity | Method | ml-1m | epinion | cosmetics | anime |
|---|---|---|---|---|---|
| $D_{cat}$ | LightGCN | 0.3852 | 0.5477 | 0.6849 | 0.3193 |
| | Multi-LightGCN | 0.3768 | 0.5454 | 0.6110 | **0.3300** |
| | CAGCN* | 0.3786 | 0.5382 | 0.6611 | 0.3206 |
| | Multi-CAGCN* | **0.4182** | **0.6667** | **0.7639** | **0.3573** |
| $D_{emb}$ | LightGCN | 0.3189 | 0.2871 | 0.4271 | 0.5134 |
| | Multi-LightGCN | **0.3206** | **0.3338** | **0.3781** | 0.4557 |
| | CAGCN* | 0.3934 | 0.3292 | 0.3833 | 0.4259 |
| | Multi-CAGCN* | **0.5229** | **0.3987** | **0.4919** | 0.4009 |

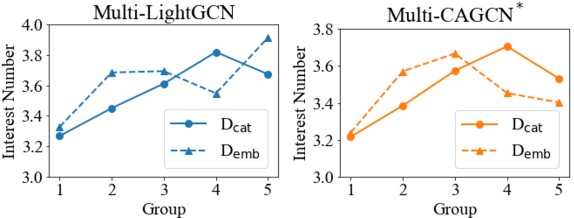

Figure 7: Average interest number for each group on ml-1m.

to user preferences to some extend. However, the trend for the last two groups is not consistent, which leave us a future direction to explicitly assign interest number based on user interest diversity in addition to the current implicit way.

## 4.5 Recommendation Diversity (RQ4)

We measure the diversity of the recommended item sets. The results are presented in Table 5 based on two diversity metrics: $D_{cat}$ in Eq.(1) and $D_{emb}$ in Eq.(2). First, the cosmetics dataset, which has the highest number of categories among the datasets, consistently exhibits the greatest diversity in comparison to the other datasets. Second, CAGCN* has a slightly reduced $D_{cat}$ than Light-GCN. This is attributed to CAGCN*'s mechanism: it assigns higher pre-computed topological-based weights to neighbors that are more densely connected to the center node (i.e., nodes that are topologically more similar). While certain nodes gain emphasis, others get overshadowed. This reduces the likelihood of recommendations based on less-similar users, resulting in the drop in diversity. Third, multi-CAGCN* has a consistent diversity enhancement (in both $D_{cat}$ and $D_{emb}$) compared with the backbone (with enhancements in 7/8 cases). We hypothesize that CAGCN* learns more accurate user interests and incorporating higher-quality embeddings amplifies the advantages of our multi-interest framework.

## 4.6 Sensitivity Analysis (RQ5)

There are two hyperparameters in the model: the number of interests and the number of hops. From Fig. 8, we draw the following observations. A larger interest number could contribute to the utility performance but not necessary maintain a higher performance. This could be due to the increasing learning difficulty and overfitting risk. Our multi-model prefers a smaller hop since (1) the multi-interest representation layer in Sec. 3.3 aggregates neighborhood information, serving as an implicit hop; (2) more layers would result in a higher level of smoothness which hides the diversity.

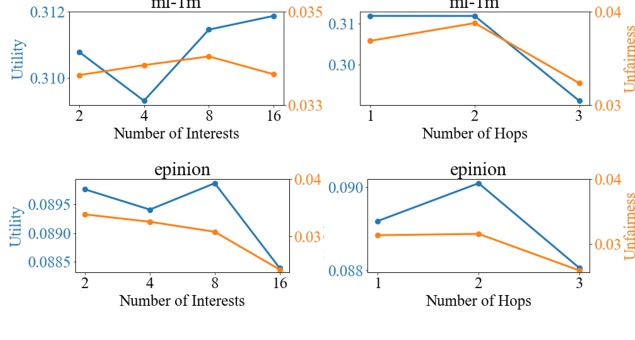

**Figure 8: Sensitivity analysis on Multi-LightGCN.**

## 5 RELATED WORKS

### 5.1 Fairness in Recommender Systems

The majority RS development is concentrated predominantly on utility performance enhancement. However, emergent concerns regarding the equitable treatment of diverse user groups have motivated the advent of fairness-aware recommender systems [16, 28, 34]. Researchers have divided users into groups and investigated the group-level unfairness based on various criteria which can be summarized into two primary categories [34]: (1) *explicit features*, which involve sensitive features such as gender [5, 32], race [10, 35] and age [8]; (2) *implicit features*, which are extracted from interactions such as the number of interactions (i.e., degree) and the amount of purchases [9, 15, 21, 31]. While the explicit features are vital to fairness discourse, they are often inaccessible due to privacy policies or users' reluctance to share such information. Consequently, our research focuses on implicit features given the profusion of user interactions in recommendation scenarios. Despite the significance of all previously mentioned features, our study explores a novel perspective within the realm of implicit features called *user interest diversity* considering its close relationship with the RS goal and its high relevance to real-world applications. Additionally, while most works adopt group information during training [16, 26], recent works have also explored group-agnostic directions with the assumption that group partitions are not available during the optimization [11, 13]. In this work, we follow this setting considering there are various ways to divide users into groups. Our goal is to develop a model that upholds fairness across diverse group divisions rather than catering to specific partition.

### 5.2 Multi-interest Recommender Systems

The main idea of multi-interest solutions is that single embedding is insufficient to represent the node's features, hence necessitating the deployment of multiple embeddings. This idea has been extensively employed in session-based recommendations - depending on how the interests are obtained, the solutions fall into attention-based and category-based methods. The attention-based methods extract interests from the interactions into interest embeddings based on the attention mechanism. MIND [14] initializes the effort to extract interests based on dynamic capsule routing. After that, ComiRec [2] leverages self-attention to learn multiple interests

given item interactions. While ComiRec [2] considers the item-to-interest relationship, Re4 [33] models interest-to-item relationship by adding regularizations. The cluster-based methods perform clustering on the interacted items and obtain representative embedding per cluster to depict interests. PinnerSage [19] clusters interacted items with the Ward hierarchical clustering method [30], and utilizes the embedding of the center item, which minimizes the sum of distance to other items within the cluster, to depict user's interests. MIP [23] assigns each interest as the representation of the latest interacted item in each cluster. In addition, MIP learns the weight to represent the preference over each interest and integrates it into the relevance score.

Beyond their application within RSs, multi-interest idea has also been applied in other representation learning tasks. For instance, the multi-interest-based random walk [20] assigns each node a target embedding along with multiple context embeddings. Similarly, in the multi-interest-based Graph Neural Network (GNN)[3], each node is characterized by several embeddings and an additional membership embedding that signifies the association with each interest. The principle of node partitioning[6] has also been adapted to accommodate multi-interest strategies, where a node is divided into several virtual nodes based on neighborhood structure, whose embeddings represent the original node.

In contrast, we delve into multi-interest in direct recommendation, emphasizing the importance of learning both user and item embeddings. Notably, in contrast to numerous studies [3, 6, 20] that increase parameter size for user representation, we employ shared global interest parameters for all users. This approach allows us to compute virtual interests in a parameter-efficient manner.

## 6 CONCLUSION

In this study, we examine whether users with varied levels of interest diversity are treated similarly/fairly in recommendation systems. Initial findings reveal a consistent disparity among user groups across different models, datasets, diversity metrics, and group partitions. This indicates the existence of *User Interest Diversity Unfairness*. Specifically, users with a broader range of interests often receive lower-quality recommendations, which has a negative impact on the user fairness and overall utility. Delving into the embedding space, we notice a trend linking group embedding alignment and utility performance. This suggests that a single embedding may not adequately represent diverse interests. To address this, we introduce a multi-interest framework where users are characterized by multiple (virtual) interest embeddings. Evaluation on two representative recommendation system backbones demonstrates that our approach better balances fairness and utility. Additionally, the learned embeddings have higher-quality and more balanced alignment in the embedding space. The proposed framework also provides more diverse recommendations. In future research, we aim to enhance the interest generation component. Currently this component is based on Softmax attention, other attentions or generative methods can be used to derive interest embeddings. For instance, we can incorporate text information and leverage large language models (LLM) for interest extraction/generation [4]. The trade-off between alignment and uniformity within the realm of multi-interest also merits investigation.

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

**Table 6: Performance (NDCG) on LightGCN backbone (The best is highlighted in bold and the runner-up is underlined).**

| Method | ml-1m | | epinion | | cosmetics | | anime | | |
|---|---|---|---|---|---|---|---|---|---|
| | NDCG↑ | Unfairness↓ | NDCG↑ | Unfairness↓ | NDCG↑ | Unfairness↓ | NDCG↑ | Unfairness↓ | Avg Rank↓ |
| LightGCN | 0.2335 | **0.0133**/0.0307 | 0.0462 | **0.0166**/0.0153 | 0.1154 | 0.0643/0.1021 | 0.2594 | 0.0154/0.0513 | 2.33 |
| DRO-LightGCN | **0.2368** | 0.0139/**0.0292** | **0.0473** | 0.0192/0.0188 | 0.1158 | 0.0670/0.1076 | 0.2553 | **0.0141**/0.0514 | 2.75 |
| ARL-LightGCN | 0.2258 | 0.0138/0.0324 | 0.0438 | 0.0167/0.0178 | 0.1063 | 0.0607/0.0897 | 0.2466 | 0.0156/0.0483 | 3.00 |
| Multi-LightGCN | 0.2363 | 0.0137/0.0499 | 0.0464 | 0.0184/**0.0093** | **0.1373** | **0.0592**/0.0756 | **0.2852** | 0.0176/**0.0471** | **1.92** |

**Table 7: Performance (NDCG) on CAGCN* backbone (The best is highlighted in bold and the runner-up is underlined).**

| Method | ml-1m | | epinion | | cosmetics | | anime | | |
|---|---|---|---|---|---|---|---|---|---|
| | NDCG↑ | Unfairness↓ | NDCG↑ | Unfairness↓ | NDCG↑ | Unfairness↓ | NDCG↑ | Unfairness↓ | Avg Rank↓ |
| CAGCN* | 0.2382 | 0.0138/0.0249 | **0.0492** | 0.0196/0.0143 | 0.1288 | 0.0692/0.1027 | 0.2619 | 0.0164/0.0521 | 2.33 |
| DRO-CAGCN* | **0.2418** | 0.0144/**0.0246** | 0.0483 | 0.0197/0.0148 | 0.1296 | 0.0708/0.1025 | 0.2613 | **0.0161**/0.0554 | 2.67 |
| ARL-CAGCN* | 0.2295 | **0.0126**/0.0318 | 0.0471 | **0.0183**/0.0180 | 0.1218 | **0.0640**/0.0899 | 0.2591 | 0.0185/0.0556 | 3.00 |
| Multi-CAGCN* | 0.2352 | 0.0132/0.0592 | 0.0480 | 0.0192/**0.0107** | **0.1472** | 0.0663/**0.0704** | **0.2875** | 0.0168/**0.0446** | **2.00** |

## A  GROUP-LEVEL EMBEDDING ALIGNMENT

In this section, we present the group embedding alignment on Light-GCN and CAGCN* on the other three datasets in Fig. 9. Generally, the trend is similar to the one in Fig. 3. When interest diversity increases, the embedding alignment shows a poor performance. The trend when using $D_{emb}$ as diversity metric is more consistent.

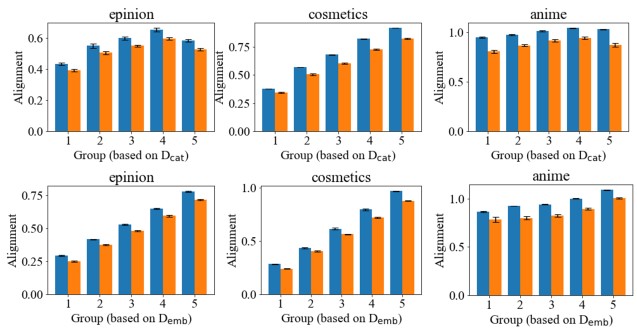

**Figure 9: Group-level embedding alignment on epinion, cosmetics, and anime datasets based on LightGCN and CAGCN*.**

## B  BEST HYPERPARAMETERS

For each model, we tune the number of hops within $\{1, 2, 3\}$. Additionally, for DRO-based model, we tune the hyperparameter $\eta$ within $\{0.0, 0.2, 0.4, 0.6, 0.8, 1.0\}$. For our model, we tune interest number within $\{2, 4, 8, 16\}$.

Best hyperparameters for each model on three seeds are as follows (the order of four datasets is ml-1m, epinion, cosmetics, and anime):

(1) LightGCN as backbone:

- **LightGCN** (number of hops): [3,3,2], [3,3,3], [2,2,2], [3,3,3].
- **DRO** (number of hops): [2,2,2], [3,3,3], [2,2,2], [2,2,2], [2,2,2]; ($\eta$): [0.6,0.6,0.6], [0.6,0.6,0.6], [0.0,0.0,0.0], [0.6,0.6,0.6].
- **ARL** (number of hops): [2,2,2], [2,3,3], [3,2,2], [2,3,2].

- **Multi** (number of hops): [2,1,1], [2,2,2], [2,2,2], [1,1,1]; (number of interests): [16,8,4], [8,4,8], [4,16,16], [2,4,2].

(2) CAGCN* as backbone:

- **CAGCN*** (number of hops): [3,3,3], [3,3,3], [3,3,3], [1,1,1].
- **DRO** (number of hops): [3,3,3], [3,3,3], [3,3,3], [1,2,1]; ($\eta$): [0.6,0.6,0.6], [0.0,0.0,0.4], [0.0,0.0,0.0], [0.6,0.6,0.6].
- **ARL** (number of hops): [3,3,2], [3,3,3], [3,3,3], [2,2,2].
- **Multi** (number of hops): [1,1,1], [2,2,2], [2,2,2], [1,1,1]; (number of interests): [16,8,4], [16,8,2], [4,16,8], [8,2,2].

## C  FAIRNESS AND UTILITY TRADE-OFF

In this section, we report another utility performance NDCG and its corresponding fairness metric (i.e., the standard deviation of group NDCG performance) in Table 6 and Table 7 based on two backbones. Note that the models are the same as in Table 3 which are selected based on the best recall value. Similar to Table 3, our proposed multi-interest framework has the highest rank among all compared methods, indicating its effectiveness in balancing fairness and utility performance.

