# OpenReview forum: "Can One Embedding Fit All? A Multi-Interest Learning Paradigm Towards Improving User Interest Diversity Fairness"
_ACM.org/TheWebConf/2024/Conference — TheWebConf24_

### Official Review · Reviewer_9Rwp · 2023-11-23

**Novelty:** 4
**Technical Quality:** 4

**Review:**

**Summary:**

The paper studies the performance of recommender systems on users with multiple interests, focusing on the fairness for this users. The paper is motivated by ensuring fairness for users with multiple interests, which differs from classical group fairness objectives because the group membership is implicit. The main contributions of the paper are (1) empirically showing that existing recommendation algorithms perform poorly on users with multiple interests, (2) designing a new framework to improve performance on these users, and (3) empirically evaluating the framework.

The paper measures interest diversity based on two metrics. The first metric is the diversity of the item categories shown in user’s historical interactions (to be used when category information is available). The second metric is diversity of interests in terms of the item embeddings (as measured by inner product) shown in user’s historical interactions (to be used when category information is unavailable).

The framework to improve performance of these users is based on the following key idea: assign a user multiple embeddings reflecting the multiplicity of interests, in addition to a center embedding reflecting the user’s key characteristics. The score function is adjusted to reflect the maximum score with respect to any of the interests. The paper provides an empirical evaluation of LightGCN and CAGCN* modified according to the multi-interest framework on the ml-1m, epinion, cosmetics, and anime dataset, and compares with DRO and ARL as baselines. The paper shows cases where the proposed framework achieves better fairness-utility tradeoffs than these baselines.

**Strengths:**
- The problem of users having multiple interests is well-motivated, and the finding that typical recommendation algorithms perform poorly on these users is practically relevant and interesting.
- The idea of introducing multiple embeddings for users to reflect different interests is natural and well-motivated.
- For baselines, in addition to the unmodified LightGCN and CAGCN*, the paper considers modified versions of LightGCN and CAGCN* with other fair baselines (DRO and ARL) that are group agnostic. This provides a nuanced comparison and helps clearly evaluate the proposed framework.

**Weaknesses:**
- In the framework, both the users and the items have multiple embeddings. It is not clear why the items also need to have multiple virtual embeddings (and not just the users)—see question below.
- While the paper frames the goal as ensuring that users with many interests are treated as a *fairness* problem, it is not clear fairness for the group of users with varied interests is a first-order goal in practice (see question below). In particular:
    - The authors do provide the following example on p.1: “While homosexual and heterosexual users have more specific preferences related to gender interests, bisexual users might exhibit a broader range of interests.” However in this example, on a dating app, the platform might request this axis of the users’ sexual orientation and explicitly ensure fairness for this protected group using standard fairness approaches for “explicit sensitive attributes”.
- That being said, performing well on users with varied interests seems well-motivated as an approach to improve utility across the whole population (as the paper examines in Section 4.1). However, the empirical evaluation of the paper in Table 3 suggests that the proposed method does not outperform existing approaches on the basis of accuracy (recall) alone.

**Questions:**

Why do the items also need to have multiple virtual embeddings?

Please discuss the fairness motivation in greater detail. Could the authors provide more real-world examples where a worse recommendation quality for users with varied interests might lead to *fairness* concerns from a practical perspective?

**Reviewer Confidence:**

3: The reviewer is confident but not certain that the evaluation is correct

**Scope:**

4: The work is relevant to the Web and to the track, and is of broad interest to the community

---

### Official Review · Reviewer_Xizy · 2023-11-24

**Novelty:** 5
**Technical Quality:** 6

**Review:**

### Summary
This paper tackles the issue that in recommender systems some users are harder to model than others. Specifically, some users have a single area of interest while others are interested ina wide, diverse array of areas. Many current recommender systems give recommendations of poor quality for users of diverse interests. To mitigate this issue, the authors present a model that learns multiple user embeddings for each user where each embedding corresponds to a different area of interest. The paper presents a motivating example, introduces measures of interest diversity as well as their model architecture, and provides thorough experimental results. Overall I find this to be a well-written and thorough paper targeting an important issue.

### Pros
* The problem is well-defined and Figure 2 provides good motivation for the rest of the paper.
* I found that the model architecture, though complex, was clearly presented.
* The experiments consider not only the accuracy vs fairness tradeoff but also analyze other properties specific to the algorithm (alignment, interest matching, recommendation diversity).

### Cons
* It is left as future work but I think the paper would benefit from a deeper analysis of the Interest Matching limitations. Given that each user has k virtual embeddings regardless of interest diversity, for users that have undiverse interests do the superfluous virtual embeddings harm recommendation quality?

* The majority of the introduction to the paper focuses on multi-interest from the user perspective. However, the model also learns virtual embeddings for each item. This appears detached from the initial motivation and it is unclear why the item virtual embeddings are needed.

### Miscellaneous
* In Equation 5, the $E_C^l[v]$ should be $E_C^l[v_n]$

**Questions:**

* Building on the first con, do the authors believe that the multi-interest framework hurts users who have singular interests?

* Regarding the second con, are the item virtual embeddings needed for the current results?

* For the motivating example in Figure 2, is it possible to verify whether the users who have a high-interest diversity in the training set also have a high-interest diversity in the test set? This would alleviate the confounding explanation that for user $i$ the set of interests expressed in the training set is mutually exclusive from the set reflected in the test set i.e. the cause of performance decrease is more directly connected to interest diversity instead of interest drift.

* For the sum in the BPR loss in 3.4, is the sum over all pairs of positive and negative items? If so does the number of terms in the sum become prohibitively large for sparse datasets and would negative sampling be needed in practice?

**Reviewer Confidence:**

3: The reviewer is confident but not certain that the evaluation is correct

**Scope:**

3: The work is somewhat relevant to the Web and to the track, and is of narrow interest to a sub-community

---

### Official Review · Reviewer_8Mqa · 2023-11-24

**Novelty:** 4
**Technical Quality:** 5

**Review:**

Summary: The paper studies whether users with diverse interests are treated fairly in recommendation systems. First, the paper shows that users with more diverse interests receive poorer recommendations in recommender systems. Second, the paper proposes an alternative method for representing user interests, in which each user is represented by multiple embedding vectors. Third, the paper evaluates this multiple-embedding system on a series of datasets and reports higher fairness and utility.

Strengths:
- Considering the impacts of recommender systems on different groups of users is an important topic and this problem seems well-motivated.

Suggestion:
- I think Figure 5 could be improved to more clearly communicate the set-up. For example, I found Figure 4 very clear, but Figure 5 had me lost.

Weaknesses:
- I found the experimental results in Table 3 a little hard to contextualize. A caption for Table 3 to orient the reader would also be helpful. I've added more questions for table 3 below.

**Questions:**

My understanding is that Table 3 compares RS backbones with and without the multi-interest framework. I have a few questions about the interpretation of these results.
- What is the number of parameters for each model with and without the multi-interest framework? I am wondering whether improvements in performance are due to the specific architecture of the multi-interest framework or to differences in the sizes of the models.
- What does the "-" bar in Table 3 refer to?
- Are the results in Table 3 statistically significant? I am wondering how to contextualize the differences in performance between different models.

**Reviewer Confidence:**

2: The reviewer is willing to defend the evaluation, but it is likely that the reviewer did not understand parts of the paper

**Scope:**

3: The work is somewhat relevant to the Web and to the track, and is of narrow interest to a sub-community

---

### Official Review · Reviewer_qDwd · 2023-12-01

**Novelty:** 6
**Technical Quality:** 7

**Review:**

My overall understanding of this work:
Starts by identifying and defining a source of disparity (user "interest diversity"/broadness and how users with broad interests overall receive poorer quality recommendations), shows how this disparity exists within current graph-based models for multiple datasets, and proposes and tests a new multi-interest framework that can be used with current graph-based model outlines (using center and virtual embeddings together instead of only a single embedding for items+users) that is designed to address this disparity.

Pros:
- Originality/Significance: This is a neat problem and it's glad to see an approach suggested!
- Clarity: Nice notation table and general clarity in defining functions.
- Clarity: Figure 4 is a clear summary of the "one embedding is not good enough" motivation. (However, it also caused some confusion - on first read I missed the fact that your framework is designed to allow center and virtual embeddings for both users and items, because Figure 4 seemed to imply that you would work with center/virtual embeddings for users only.)
- Quality: The research questions you have outlined seem like a very neat breakdown of the important points, especially with RQ3 through RQ5 exploring additional benefits on top of simple alignment and performance tests. (I'm curious how the future work for these last three RQs will go.)

Cons:
- Clarity: I'm confused what the (A), (B), (C) label texts in Figure 2 are supposed to mean (outside of them being used as LaTeX reference anchors). It looks like all the graphs are showing model utility/performance across a range of models, datasets, and across multiple user subgroups on the X-axis split different ways for each subfigure, so the texts are just creating extra confusion.
- Clarity: This may be GCN-specific context - I'm still lost on what a useful high-level descriptor of the global interest embeddings would be. They aren't described in detail anywhere in this paper.
- Clarity: Figure 5, the "center embedding aggregator" focus area is very confusing (see questions)
- Quality: Not sure why the "multi-interest framework works better with large dataset" performance comparison focuses first on dataset size vs. category count or domain-specific usage details. I'm curious to see more about how diversity of items and categories in this dataset affects results, instead of only the overall dataset size.

Specific suggestions:
- Clarity: Section 3.1 (source of unfairness) may tie better into section 2 (exploring preliminary results + building a motivation) as a way of wrapping it up and providing transition motivation. Currently section 3 seems like it's supposed to introduce your new framework but 3.1 only goes back to talking about previous preliminary performance and motivations.

Context for my review:
I have generally surveyed (focusing on motivations and application challenges, not technical implementation) various works in fairness in recommender and classification systems, largely those focused on fairness for users across distinct identity groups and some work with ranking fairness. I am not familiar with the methods of LightGCN or CAGCN, and I am still unsure how exactly your framework ties into these backbones after reading the paper or what backbone-specific details I am lacking the context for (e.g. I'm not sure what "number of hops" as discussed in RQ5 means exactly and am assuming this is part of the GCN functionality outside of your new framework). This makes judging originality and significance a bit trickier too, so most of my opinion is based on quality and clarity.

**Questions:**

If I try to summarize the framework based on Figure 5 right now, I'm understanding it as:
1. Input: a user-item interaction graph, with some prebuilt / seed embeddings for each user and each item (the original center embeddings)
2. Some part of the representation layers (the interest embedding generator): for each user's item interaction history, apply some number of global interest embeddings (attention? grouping similar items together? rough binning of recognized interest types?) to generate a set of user virtual embeddings, which essentially describe a set of interests for each user.
3. Some part of the representation layers (the center embedding aggregator): for each user's item interaction history, identify some set of item virtual embeddings with one for each item (where do the item virtual embeddings come from? are these also generated with the interest embedding generator, just with the user/item positions flipped? how is this exact set selected (I did see the process described in text, but the figure is confusing)?) and do something with them. Based on the figure and reading from top down, it looks like you are generating virtual embeddings based on this selection, but based on the heading and text description it looks like the point is to generate an updated user center embedding.
4. After some number of layers, end up outputting a set of updated center embeddings for each user+item and multiple sets of virtual embeddings for each user and item (possibly of varying count, the figure doesn't make this clear but the text mentions this).

Is this broadly accurate? (if yes, most of my clarity complaints in the review are still there because Figure 5 did take a while to parse.)

**Reviewer Confidence:**

2: The reviewer is willing to defend the evaluation, but it is likely that the reviewer did not understand parts of the paper

**Scope:**

3: The work is somewhat relevant to the Web and to the track, and is of narrow interest to a sub-community

---

### Decision · Program_Chairs · 2024-01-22

**Decision:**

Accept

**Comment:**

Our decision is to accept. Please see the AC's review below and improve the work considering that and the reviewers' feedback for cemera-ready submission.

"This paper studies the question of fairness in item recommendation - specifically, the authors consider "fairness" as giving similar levels of utility to users who have a) more specific, narrow preferences and b) users who have more variable/wide types of preferences. The authors achieve this goal by allowing for multiple embeddings representing different types of interest.

 Overall the reviewers appreciated that the paper was well-written in general (although note a few points where clarity could be improved, such as part of Figures 2 and 5, and certain key terms), and appreciated the importance of the problem. Some reviewers raised concerns about how this new approach compares with benchmarks: for example, Table 3 shows that their proposed method only sometimes outperforms benchmarks on recall. In response, during the rebuttal phase the authors reran the experiments with more simulations and showed that their approach has the highest rank (tradeoff between fairness and accuracy). Also during the rebuttal phase, the authors explored datasets with larger numbers of items and categories, exploring distribution shift between the training and test dataset.

 In one more minor point, I think that the dating example used by the authors isn't ideal - at least two reviewers mentioned this as suboptimal (one informally in an offline conversation). I've added a note to the authors about this example in a separate comment, but I think this doesn't substantively affect my opinion of the main contribution of the paper."